# Aggregation of Mouse Serum Amyloid A Protein Was Promoted by Amyloid-Enhancing Factors with the More Genetically Homologous Serum Amyloid A

**DOI:** 10.3390/ijms22031036

**Published:** 2021-01-21

**Authors:** Xuguang Lin, Kenichi Watanabe, Masahiro Kuragano, Kiyotaka Tokuraku

**Affiliations:** 1Muroran Institute of Technology, Graduate School of Engineering, Muroran 050-8585, Japan; linxuguang4000@163.com (X.L.); gano@mmm.muroran-it.ac.jp (M.K.); 2Research Center of Global Agromedicine, Department of Veterinary Medicine, Obihiro University of Agriculture and Veterinary Medicine, Obihiro 080-8555, Japan; knabe@obihiro.ac.jp

**Keywords:** amyloid A amyloidosis, amyloid enhancing factor, homology, serum amyloid A, quantum-dot

## Abstract

Amyloid A (AA) amyloidosis is a condition in which amyloid fibrils characterized by a linear morphology and a cross-β structure accumulate and are deposited extracellularly in organs, resulting in chronic inflammatory diseases and infections. The incidence of AA amyloidosis is high in humans and several animal species. Serum amyloid A (SAA) is one of the most important precursor amyloid proteins and plays a vital step in AA amyloidosis. Amyloid enhancing factor (AEF) serves as a seed for fibril formation and shortens the onset of AA amyloidosis sharply. In this study, we examined whether AEFs extracted and purified from five animal species (camel, cat, cattle, goat, and mouse) could promote mouse SAA (mSAA) protein aggregation in vitro using quantum-dot (QD) nanoprobes to visualize the aggregation. The results showed that AEFs shortened and promoted mSAA aggregation. In addition, mouse and cat AEFs showed higher mSAA aggregation-promoting activity than the camel, cattle, and goat AEFs. Interestingly, homology analysis of SAA in these five animal species revealed a more similar amino acid sequence homology between mouse and cat than between other animal species. Furthermore, a detailed comparison of amino acid sequences suggested that it was important to mSAA aggregation-promoting activity that the 48th amino acid was a basic residue (Lys) and the 125th amino acid was an acidic residue (Asp or Glu). These data imply that AA amyloidosis exhibits higher transmission activity among animals carrying genetically homologous SAA gene, and may provide a new understanding of the pathogenesis of amyloidosis.

## 1. Introduction

Amyloidosis is characterized by the accumulation of an insoluble β-sheet-rich structure of amyloid fibrils into plaques in extracellular spaces of different organs and tissues, causing organ dysfunction [1,2]. As of now, more than 50 different peptides or proteins have been found that are associated with amyloidosis in humans and livestock, and their fibrils lead to the pathogenesis of many disorders such as rheumatoid arthritis, tuberculosis, and others [3,4,5]. Amyloid A (AA) amyloidosis, one of the most common forms of clinically important amyloidosis, is related to chronic inflammatory diseases and chronic infections. The structure of these amyloid fibrils may affect their ability to spread to different sites in a cell and between organisms in a prion-like mechanism, but the mechanism by which amyloid fibrils form in vivo and in vitro remains largely unclear although the structure of many precursors has been elucidated [5,6,7,8].

Serum amyloid A (SAA), an evolutionarily highly conserved acute phase protein in vertebrates and invertebrates, is predominantly secreted by hepatocytes in the liver and is also produced by a variety of cells and tissues [9,10,11]. In fields of regulating inflammation and immunity and lipid metabolism, SAA protein plays an important role that benefits the body [6,11,12]. SAA is an important amyloid precursor and plays a key role in AA amyloidosis which impacts ∼1% of patients with chronic inflammation [13,14]. Furthermore, SAA is a clinically important biomarker for inflammatory diseases [15,16]. The additional administration of an amyloid enhancing factor (AEF) which can be the nucleate of fibril formation in vitro, has been shown to shorten the onset of AA amyloidosis markedly from months to days, while amyloid fibrils themselves act as an AEF (Figure 1) [17,18,19,20,21]. Oral administration and subcutaneous injection of AEF from different animal species with silver nitrate can induce AA amyloidosis in mice, indicating a prion-like transmission [17,22,23]. In addition, AA amyloidosis has been found to spread through feces in cheetah [24]. Perhaps these substances are involved in the transmission of AA amyloidosis and play an important role in the spread of this disease. Despite this knowledge, the molecular mechanism of AEF activity in AA amyloidosis remains unclear.

Previously, we reported a real-time imaging method of various amyloid proteins such as amyloid β (Aβ_42_), tau, α-synuclein, and SAA aggregation with quantum-dot (QD) nanoprobes, which was using fluorescence probes in imaging by confocal and fluorescent microscopy, and developed a microliter-scale high-throughput screening system (MSHTS) to search for substances that exhibit aggregation inhibitory activity by applying this imaging method [6,25,26,27]. Moreover, QDs are useful for long-term, single-molecule imaging in vitro. In the MSHTS system, a sample of 5 μL in volume is needed for analysis in a 1536-well plate, and the inhibitory activity is estimated by half-maximal effective concentration (EC_50_) [6,26,28]. These imaging techniques that employ QDs serve as quick, easy, and powerful tools to apply to in vitro screening and monitoring.

In this study, we attempted to clarify whether AEFs extracted from different animal species (camel, cattle, cat, goat, and mouse) with AA amyloidosis act as seed to promote mouse SAA (mSAA) protein aggregation and their function in transmission between different animals in vitro. Therefore, we adopted a QD nanoprobe to visualize, in 2D and 3D, mSAA aggregation promoted by AEFs from different animal species. Furthermore, we analyzed the homology of SAA sequences from these five animal species and humans then established a homologous evolutionary tree by DNAMAN software to assess the relationship between gene homology and the degree of SAA aggregation. QD imaging data in vitro suggested the possibility of transmission of AA amyloidosis between different animal species.

## 2. Results

### 2.1. Real-Time Imaging of mSAA Aggregation Using QDs

In previous studies, we demonstrated that QDs can be used to observe recombinant mouse SAA (mSAA.1) aggregation under fluorescence microscopy and quantify the amount of aggregates from microscopic images [6]. In this study, 10 µM mSAA protein mixed with QDs in PBS was incubated at 37 °C in 1536-well plate for one week to induce aggregation, and 2D images were captured every 24 h by fluorescence microscopy from 0 h to 168 h, respectively (Figure 2A). The fluorescent micrographs showed that mSAA protein did not aggregate at this concentration. Figure 2B plots the amount of aggregates estimated from the standard deviation (SD) of the brightness of each pixel of a fluorescence microscope image according to our previous reports [25,26]. The SD values correlate with the amount of mSAA protein aggregates. However, the SD value did not increase after 168 h of incubation in this condition (Figure 2B). Our previous report [6] showed that SAA spontaneously aggregates with increasing concentration, reaching a maximum SD value when the concentration is 50 µM. These results suggest that mSAA cannot spontaneously aggregate at a low concentration of about 10 μM.

### 2.2. Imaging of mSAA Aggregation Promoted by AEFs from Various Animals

AEF as a seed can promote SAA protein aggregation and accelerate pathology in AA amyloidosis. In this study, we used different concentrations of AEFs (20%, 40%, 60%, 80%, and 100%) that were extracted and purified from five animal species (camel, cat, cattle, goat, and mouse). These AEFs were added to 10 μM mSAA and aggregates in those samples were visualized using QDs. After incubation for 168 h, the images at different concentrations and in different animal species were compared (Figure 3A). The results show that the AEFs of the five animals all have different degrees of promoting aggregation on mSAA protein. Control group samples (AEFs only) seldom had aggregates after 168 h of incubation. Both mouse and cat AEFs promoted mSAA aggregation with 100% AEF, but many aggregates were observed even at low concentrations of AEF, such as 20% or 40%, in the presence of mouse AEFs. AEFs from other animal species (camels, goats, and cattle) also increased aggregates as the amount of AEF added was increased, but the increase was significantly less than that of mouse and cat AEFs. We also confirmed these aggregates were β-sheet-rich amyloid fibrils by staining with thioflavin T (ThT) (Appendix A).

We then compared the time-dependent manner of mSAA aggregation in the presence of 100% AEFs using SD values (Figure 3B). The results show that the SD values increased over time in all samples to which AEFs were added, but the amount of change differed. Similar to Figure 3A, aggregation of the camel, cattle, and goat group was slower than that of the mouse and cat group. Interestingly, the SD value of the cat AEF sample was higher than that of the mouse as was its speed of increase.

### 2.3. D observation of mSAA Aggregates with Various Animal AEFs

The 3D aggregates with five animal AEF (100%) samples after incubation for 168 h in the 1536-well plate with QDs were observed and captured by the confocal microscopy directly (Figure 4A). The aggregation speed of each sample in the 3D imaging (Figure 4A) was consistent with the 2D imaging and, over time (Figure 3B), the thickness of each protein increased due to aggregates [27]. The thickness of mouse and cat samples was much greater than that of camel, cattle, and goat. The XY view images showed that the size and density of cattle, cat, camel, and goat aggregation were more similar, showing a dotted-like aggregation whereas the mouse sample aggregation showed a mesh-like form (Figure 4B).

### 2.4. Homology Analysis

Owing to SAA protein being encoded by a family of closely-related genes, it is a highly evolutionarily conserved protein in vertebrate [9]. In this study, a total of 27 SAA protein sequences had high homology (at least 77.75%) across the six vertebrate species studied. A phylogenetic tree showed that the group with mouse and cat (red box) had higher homology than the other animals and humans (Figure 5). In our previous report, we showed that the sequences of different SAA subtypes had high homology among different samples from the same individual animal but almost no differences between different subtypes within a given animal species [4]. Moreover, when these SAA sequences were compared (Figure 6), we found that in mouse and cat sequences, the 48th amino acid was Gln and Ile, whereas in cattle, goat, and camel, it was Lys with a basic side chain (red arrow). The 125th amino acid of mouse and cat was Glu with an acidic side chain, but it was Ala in the other animals (blue arrow). The differences in these amino acids may affect SAA protein aggregation because the charge on the amino acid side chains significantly affects protein-protein interactions.

## 3. Discussion

AA amyloidosis (reactive, secondary), which is characterized by the accumulation and deposition of β-sheet-rich and non-branching amyloid fibrils, associates with a number of pathological conditions in which can be associated with a severe complication of chronic inflammatory and other inflammatory-related diseases [29,30,31,32]. During the occurrence and development of this disease, SAA protein plays an important role in AA amyloidosis. In our previous research, we already showed that mouse AEF can induce and shorten AA amyloidosis [6,19,23]. Moreover, AEF consisted almost exclusively of AA-related protein [17]. There are related reports in which AEF plays an important role in the cross-species transmission of AA amyloidosis among different species [2,23,33,34]. Moreover, current research reveals that AA amyloidosis could be transmitted by a prion-like mechanism [35]. As of now, there is no successful therapy that directly and clearly targets amyloid aggregation and deposition in organs or tissues, and there are no approved treatments to revert or arrest the progression of amyloidosis [36,37]. In previous studies [6,25,27,38,39], we have demonstrated that rosmarinic acid, which is one active compound of the Lamiaceae family, shows high inhibitory activity for amyloid fibrils formation. Moreover, it has also been reported that several small-molecule compounds such as natural polyphenols suppressed amyloid fibril formation by generating small ‘‘off-pathway’’ oligomers that non-toxic to cells [40,41,42,43]. These reports provide the molecular mechanisms involved in amyloidosis and propose more efficient drugs for therapy amyloidosis.

In this study, in order to prove whether AEFs from different animal species can promote and accelerate mSAA protein aggregation in vitro, we used a QD imaging method [26]. We selected five animal species (camel, cat, cattle, goat, and mouse) with AA amyloidosis and extract AEFs from the liver or kidney. Those AEFs at different concentrations were mixed with mSAA protein and incubated in a 1536-well plate for 168 h. We found that all of these AEFs promoted mSAA protein aggregation after incubation (Figure 3). A comparison of 2D imaging data suggests that mouse and cat AEFs promote activity more than the AEFs of other animals (camel, cattle, and goat). mSAA aggregation in the presence of 100% cat AEF was faster than in the presence of 100% mouse AEF and its SD value was higher (Figure 3B). On the other hand, mouse AEFs promoted aggregation even at low concentrations (Figure 3A). 3D imaging showed that some differences exist between these aggregate forms. The aggregation induced by mouse AEFs had a mesh-like form but those induced by cat, camel, cattle, and goat AEFs were a dotted-like form. Since the morphology of the aggregates affects the SD value, the high SD value of the cat AEF sample (Figure 3B) may reflect a difference in the morphology of the aggregate rather than the amount of the aggregate [44]. In our previous research [4,19,23], we observed amyloid fibrils in AEFs, which were extracted and purified from different animal organs, by transmission electron microscopy). In that study [4], we showed that AEFs contain multiple peptides, including SAA fragments, by SDS-PAGE and Western blot analysis. An interesting future study would be to analyze the ultrastructure of mSAA co-aggregated with diverse AEFs.

SAA protein is an evolutionarily conserved protein in vertebrates with high homology among different animal species. Since AA amyloid can be transmitted between different animal species, the processes underlying amyloidosis in those species may be similar [4,34]. Species with similar homology are more likely to transmit AA amyloidosis to each other. In this study, the SAA sequences in humans and five animal species (camel, cat, cattle, goat, and mouse) showed 77.75% homology, with mouse and cat showing the most similar homology. Some reports have shown that in α-synuclein the putative prion-like templating and spreading ability of amyloid seeds greatly depend on their amyloid fibril size [45,46]. Our previous study also indicated that in different animal species the amyloid fibrils reveal high genetic homology and morphological feature similarity in fibrils width and crossover distance [4]. In this study, we found that species with similar homology have a higher ability to enhance SAA aggregation. We speculate that fibrils with similar morphology are more infectious in infectious amyloidosis.

Most importantly, the 48th amino acid of mouse and cat was Gln and Ile, but that of cattle, goat, and camel was a basic Lys residue (Figure 6, red arrow). The 125th amino acid of mouse and cat was an acidic Glu residue, but that of other animals was Ala (Figure 6, blue arrow) in the C-terminal. In mice in which amyloidosis had been induced, treatment with the C-terminal peptide inhibited further extension of amyloid fibrils in AApoA2 amyloidosis [47]. We speculate that Glu125 may affect the aggregation and transmission of SAA in different species.

In summary, we successfully visualized the aggregation of mSAA mixed with AEFs from five animal species using QD nanoprobes. Furthermore, we demonstrated that AEFs from species with similar homology enhanced SAA aggregation. This information may provide a better understanding of amyloid disease and lead to the development of novel therapies.

## 4. Materials and Methods

### 4.1. Preparation of AEFs Extracts

AEFs from different animals were extracted according to Pras’ method [48]. Mouse (*Mus musculus*), cattle (*Bos taurus*), goat (*Capra aegagrus hircus*), camel (*Camelus bactrianus*), and cat (*Felis catus*) with AA amyloidosis, which was identified by section antibody staining [4,23], were used in the present study. Animal experiments were approved by the Research Center of Global Agromedicine of Obihiro University of Agriculture and Veterinary Medicine to Obihiro, Japan (Permission No. 19-179: 7 Oct 2019). The preparation of mSAA protein (SAA1.1: accession No. NP_033143) was performed as previously described [6]. The Lowry [49] method was used to determine mSAA protein concentration of these five animal AEFs samples, which were stored at −80 °C until use.

### 4.2. Imaging of mSAA Protein

Ten µM mSAA protein sample was mixed with 30 nM QD605 (Q21501MP, Thermo Fisher Scientific, Waltham, MA, USA) in PBS. Five μL of the sample was injected into each well of a 1536-well plate and centrifuged at 3700 rpm for 5 min at room temperature (PlateSpin, Kubota, Tokyo, Japan). The plate was incubated at 37 °C in an air incubator (SIB-35, Sansyo, Tokyo, Japan), observed, and images were captured at 0 h, 24 h, 48 h, 72 h, 96 h, 120 h, 144 h, and 168 h using an inverted fluorescence microscope (TE2000, Nikon, Tokyo, Japan).

### 4.3. Imaging of AEF Enhancing mSAA Protein Aggregation

Various concentrations (20%, 40%, 60%, 80%, and values represent the relative ratio of mSAA protein) of AEFs (camel, cattle, goat, cat, and mouse) were mixed with 10 µM mSAA and 30 nM QD605 in PBS, pH 7.4. 100% AEFs, which served as the control group, were mixed with 30 nM QD605 in PBS. Then, 5 μL of the sample was transferred into a 1536-well plate, the plate was centrifuged at 3700 rpm for 5 min, then incubated at 37 °C in an air incubator. Samples were observed and images were captured at 0 h, 24 h, 48 h, 72 h, 96 h, 120 h, 144 h, and 168 h using an inverted fluorescence microscope. The amount of amyloid aggregates was estimated from fluorescent micrographs according to our previous reports [6,25,27]. At 168 h, the 3D images of 100% concentration samples of different animal samples were captured by a confocal laser microscope (Nikon C2 Plus, Nikon, Tokyo, Japan). The same angle was adjusted to contrast these 3D images, then slice images of aggregates of each animal’s 3D images were selected. Moreover, we performed thioflavin T (ThT) fluorescence observe as control, briefly, 100% of AEFs (camel, cattle, goat, cat, and mouse) were mixed with 10 µM mSAA and 50 µM ThT added in PBS and transferred into a 1536-well plate, then incubated and observed by fluorescent microscopy (TE2000, Nikon, Tokyo, Japan).

### 4.4. Homology Analysis

On the NCBI website (http://www.ncbi.nlm.nih.gov), we searched a total of 27 sequences of SAA protein from five animal species (cattle, goat, mouse, cat, and camel) and humans. Since there are many subtypes in SAA protein, we selected sequences with the same length in this study and compared their homology. DANMAN software (Lynnon Biosoft, San Ramon, CA, USA) was used to analyze the homology of these SAA protein sequences and to establish a homology evolution tree by protein multiple sequence alignment.

## Figures and Tables

**Figure 1 ijms-22-01036-f001:**
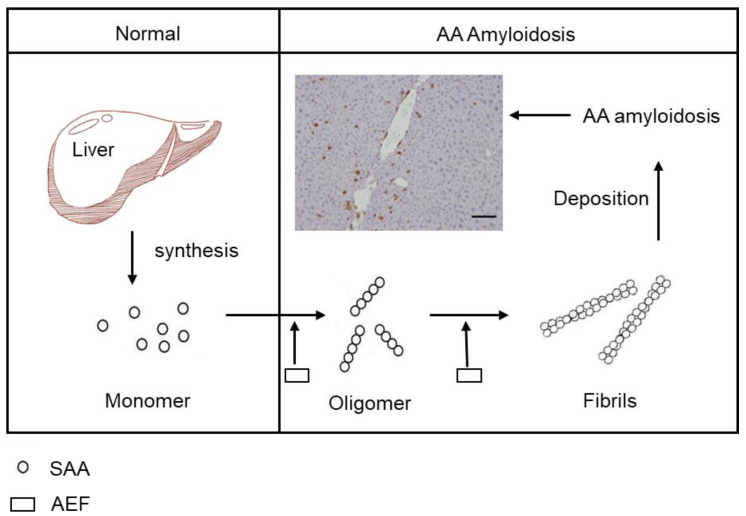
Mechanism of amyloid A (AA) amyloidosis development and enhancement by amyloid enhancing factor (AEF). Serum amyloid A (SAA) protein is synthesized in the liver in normal conditions, but in AA amyloidosis, under the stimulus of AEF, SAA protein aggregates into fibrils and is deposited in the liver. Scale bar indicates 100 μm.

**Figure 2 ijms-22-01036-f002:**
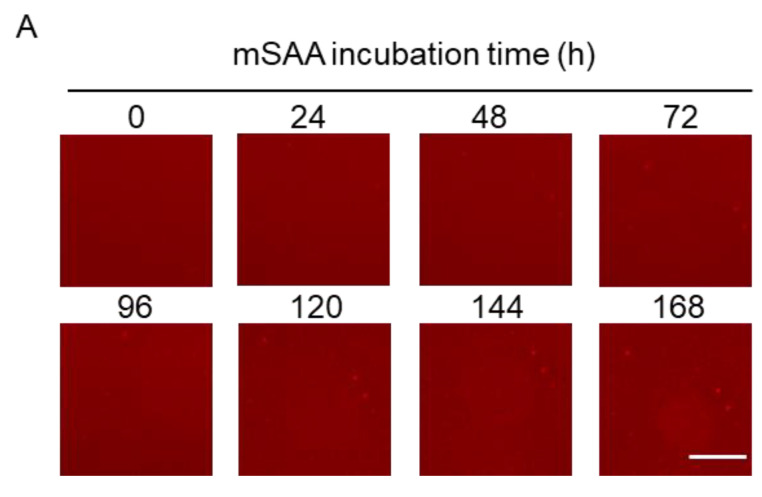
Fluorescence images of mouse serum amyloid A (mSAA) aggregation using quantum-dot (QD) nanoprobes. (**A**) 2D images of mSAA aggregations at 0 h, 24 h, 48 h, 72 h, 96 h, 120 h, 144 h, and 168 h, respectively. Scale bar in 168 h fluorescent micrograph indicates 100 μm. (**B**) SD value of each image by time-dependent mSAA protein aggregation. mSAA protein could not spontaneously aggregate at a low concentration. Each plot represents the mean ± SEM. *n* = 3 separate experiments.

**Figure 3 ijms-22-01036-f003:**
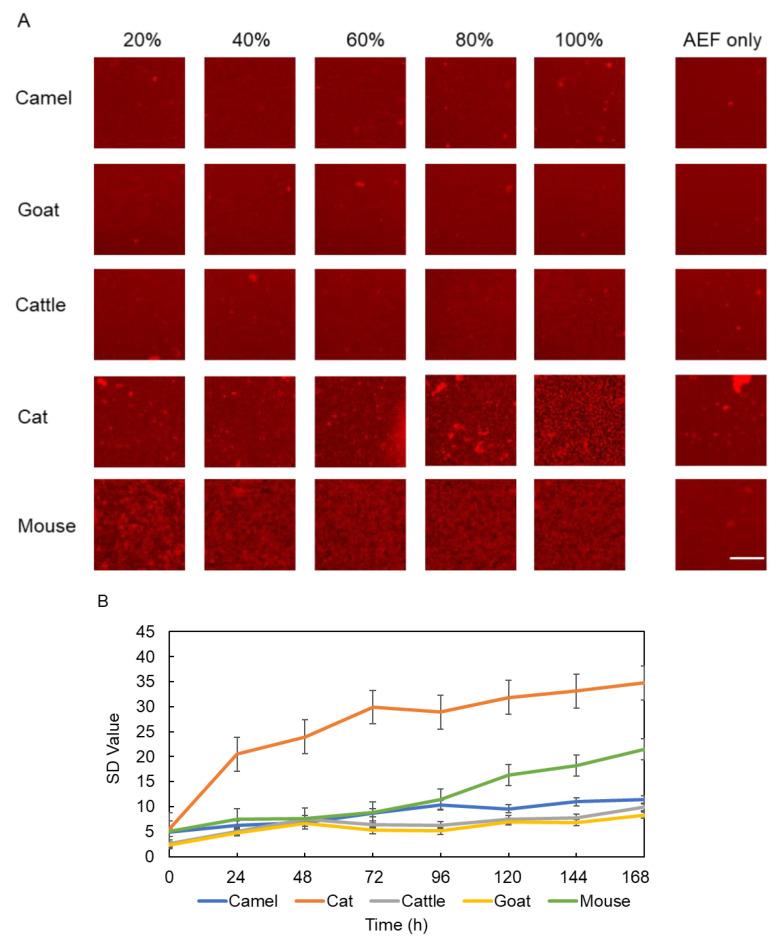
(**A**) Imaging of mouse serum amyloid A (mSAA) aggregates in the presence of 20%, 40%, 60%, 80%, and 100% amyloid enhancing factor (AEFs) after 168 h of incubation. Among the AEFs from five animals, mouse and cat AEFs showed high mSAA aggregation-promoting activity. Scale bar in fluorescent micrograph indicates 100 μm. (**B**) The temporal increase of SD values in the presence of 100% AEFs from five animal species within 168 h. The SD values increased over time in all samples to which AEFs were added. Each plotted value represents the mean ± SEM. *n* = 3 separate experiments.

**Figure 4 ijms-22-01036-f004:**
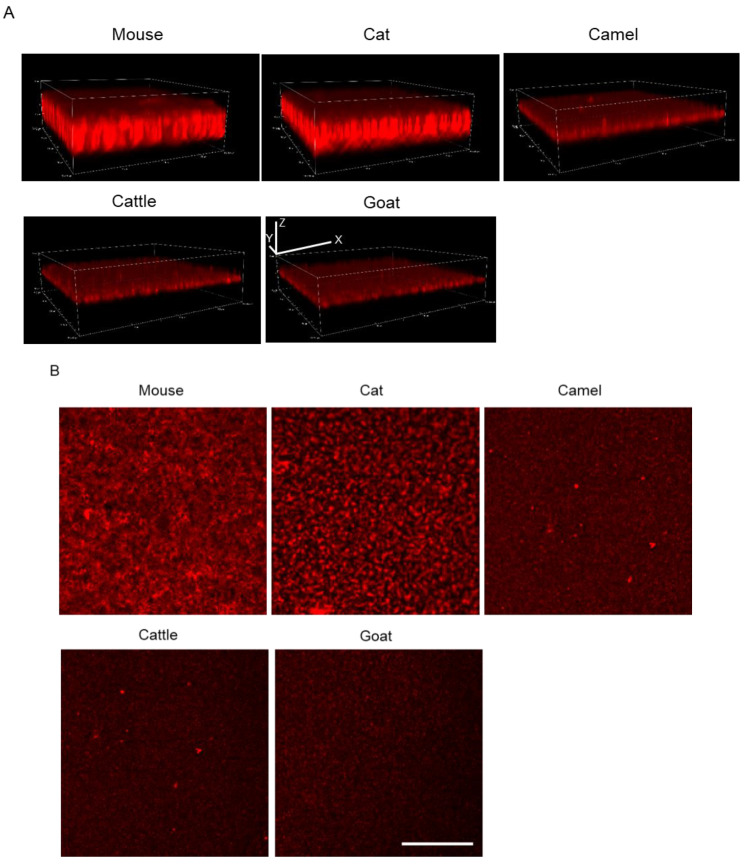
(**A**) 3D reconstruction images of the mouse serum amyloid A (mSAA) aggregates in the presence of camel, cattle, goat, cat, and mouse amyloid enhancing factor (AEFs) (final concentration = 100%) after 168 h of incubation. The thickness of aggregates in the presence of mouse and cat AEFs was much greater than that of camel, cattle, and goat AEFs. Three-dimensional white lines with the letters x, y, and z each indicates a scale of 50 μm. (**B**) Slice images of aggregates of each sample in panel (**A**). The morphology of each aggregate was different depending on the added AEF. Scale bar in the goat fluorescent micrograph indicates 100 μm.

**Figure 5 ijms-22-01036-f005:**
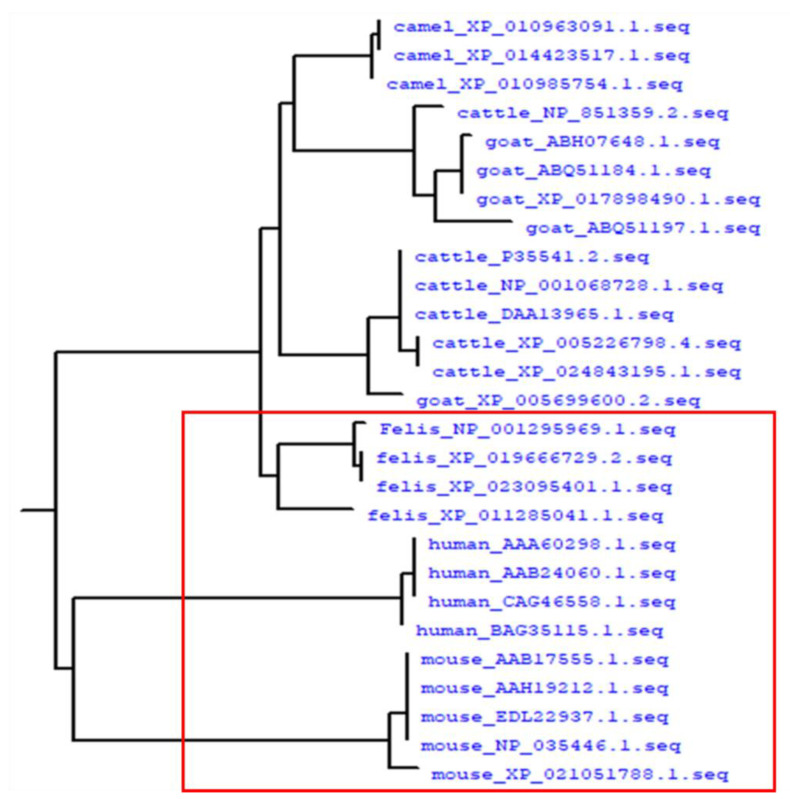
Phylogenetic tree of SAA gene sequences from camel, cattle, goat, cat, mouse, and human. The numbers following the species’ names indicate the National Center for Biotechnology Information (NCBI) accession numbers. The red box indicates clustering of sequences from mouse, human, and cat isoforms with similar homology.

**Figure 6 ijms-22-01036-f006:**
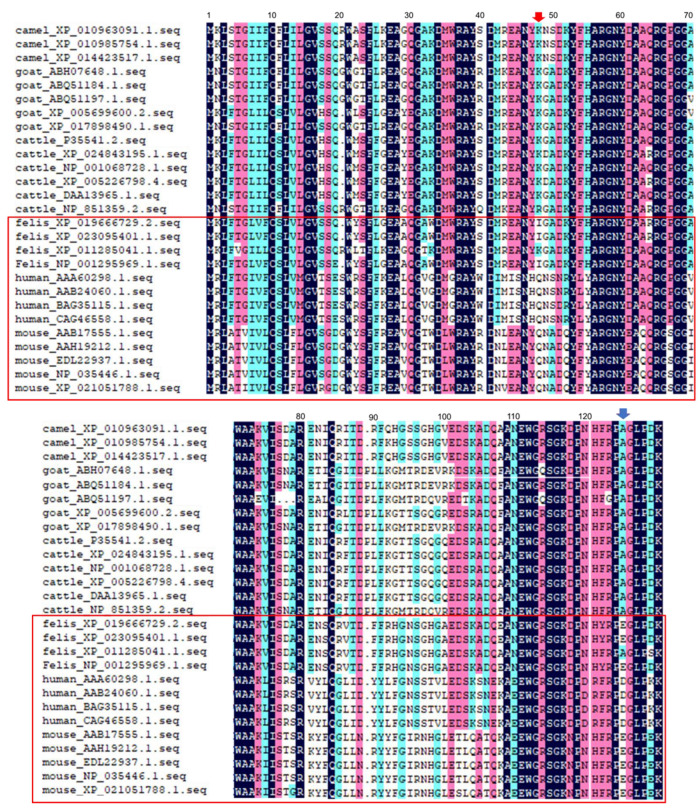
Monomeric structure of serum amyloid A (SAA) sequences from five animal species and human. Color description of SAA gene homology sequences: 100% similarity (blue); similarity > 75% and <100% (pink); similarity >5 0% and <75% (green); similarity <50% has no color. Red and blue arrows indicate the position of the 48th and 125th amino acid. Red boxes indicate clustering of sequences from mouse, human, and cat isoforms with similar homology.

## Data Availability

http://www.ncbi.nlm.nih.gov.

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
