# Peer review of "Aggregation of Mouse Serum Amyloid A Protein Was Promoted by Amyloid-Enhancing Factors with the More Genetically Homologous Serum Amyloid A"

_ijms, 2021, doi:10.3390/ijms22031036_

Round 1

Reviewer 1 Report

To the Authors:

In the present manuscript, Lin et al. claim that the amyloid enhancing factors (AEFs) from diverse animal species have similar genetic homology and transmission activity in Amyloid A (AA) amyloidosis, and their results could provide a better understanding of the pathogenesis of AA amyloidosis.

The manuscript presents original data and might be relevant not only for AA amyloidosis, but also for other proteinopathies. There are, however, several issues that must be addressed by authors, as described below.

Major Concerns:

  1. 2D images of mSAA aggregations in Fig. 2a have poor resolution/quality. Authors should provide better images.
  2. The putative prion-like templating and spreading ability of amyloid seeds greatly depend on their size (PMID: 28576704; PMID: 31254094). The authors should acknowledge these studies in their manuscript, and provide a quantification of the size of AEF-induced mSAA aggregates for the different species.
  3. Amyloid fibrils bind a wide variety of aggregate-specific dyes, e.g ThT, K114, Congo Red. In parallel to the morphological time-lapse imaging of mSAA aggregation using QD nanoprobes, the authors should follow AEF-induced mSAA aggregations using one, or more, of such dyes.
  4. Several small-molecule compounds have been shown to re-direct the aggregation cascade of several amyloid proteins into off-pathway amorphous aggregates that are non-toxic to cells (PMID: 21740906; PMID: 29124175; PMID: 20385841; PMID: 30875761). The authors should acknowledge these studies in their manuscript, and put them into perspective with their own hypothesis.

Minor concerns:

- The title is too long. It should be shortened.

Author Response

In the present manuscript, Lin et al. claim that the amyloid enhancing factors (AEFs) from diverse animal species have similar genetic homology and transmission activity in Amyloid A (AA) amyloidosis, and their results could provide a better understanding of the pathogenesis of AA amyloidosis.

The manuscript presents original data and might be relevant not only for AA amyloidosis, but also for other proteinopathies. There are, however, several issues that must be addressed by authors, as described below.

Thank you for your good evaluation. We revised the paper according to the opinions.

Major Concerns: 

2D images of mSAA aggregations in Fig. 2a have poor resolution/quality. Authors should provide better images.

We have changed the Fig. 2a please check.

The putative prion-like templating and spreading ability of amyloid seeds greatly depend on their size (PMID: 28576704; PMID: 31254094). The authors should acknowledge these studies in their manuscript, and provide a quantification of the size of AEF-induced mSAA aggregates for the different species.

Following the reviewer's comment, we added a description (lines 224-230).

Amyloid fibrils bind a wide variety of aggregate-specific dyes, e.g ThT, K114, Congo Red. In parallel to the morphological time-lapse imaging of mSAA aggregation using QD nanoprobes, the authors should follow AEF-induced mSAA aggregations using one, or more, of such dyes.

According to the reviewer's comment, we confirmed whether the AEF-induced mSAA aggregates were amyloid fibrils by staining with ThT, which is an amyloid fibril-specific dye. The fluorescence microscopic images of the aggregates that stained with ThT was shown in Supplementary Figure 1, and the explanation was added in the text (lines 118-120 and lines 274-277).

Several small-molecule compounds have been shown to re-direct the aggregation cascade of several amyloid proteins into off-pathway amorphous aggregates that are non-toxic to cells (PMID: 21740906; PMID: 29124175; PMID: 20385841; PMID: 30875761). The authors should acknowledge these studies in their manuscript, and put them into perspective with their own hypothesis.

Following the reviewer's comment, we added a description (lines 190-196).

Minor concerns:

 - The title is too long. It should be shortened.

We shortened the title to “Aggregation of mouse serum amyloid A protein was promoted by amyloid-enhancing factors with the more genetically homologous serum amyloid A”.

Reviewer 2 Report

The authors examined the effect of amyloid enhancing factor (AEF) on aggregation of mouse serum amyloid A protein in vitro using quantum-dot nanoprobes. AEFs from mouse and cat promoted aggregation of mouse serum amyloid A more intensively than those from camel, cattle, and goat. Homology analysis of serum amyloid A in these five animal species revealed a more similar amino acid sequence homology between mouse and cat than between other animal species.

This is an interesting study providing important insights into current knowledge on the pathogenesis of amyloidosis. The manuscript is well written, and I do not have any critical comments.

Minor issues to strengthen this manuscript are raised as follows: 

  1. The authors mentioned in the last sentence of abstract that “These data imply that the AEFs from different animal species have similar genetic homology and transmission activity in AA amyloidosis”. Although analyses of amino acid sequence suggested genetic homology to some extent, transmission activity seemed to be different among 5 species. I would suggest modifying the conclusion from this viewpoint.
  2. Abbreviations used in figures should be spelled out in figure legends to increase the readability for non-experts. These include SAA and AEF in Figure 1, mSAA in Figure 2, mSAA and AEFs in Figure 3 and 4, SAA and NCBI in Figure 5, and SAA in Figure 6.
  3. Findings obtained from Figures 2, 3, and 4 should be described in the legends for these figures. It will facilitate comprehension.

Author Response

 The authors mentioned in the last sentence of abstract that “These data imply that the AEFs from different animal species have similar genetic homology and transmission activity in AA amyloidosis”. Although analyses of amino acid sequence suggested genetic homology to some extent, transmission activity seemed to be different among 5 species. I would suggest modifying the conclusion from this viewpoint.

Following the reviewer's comment, we modified the last sentence of abstract (lines 25-26).

Abbreviations used in figures should be spelled out in figure legends to increase the readability for non-experts. These include SAA and AEF in Figure 1, mSAA in Figure 2, mSAA and AEFs in Figure 3 and 4, SAA and NCBI in Figure 5, and SAA in Figure 6.

Following the reviewer's comment, we added the abbreviated description in each figure.

Findings obtained from Figures 2, 3, and 4 should be described in the legends for these figures. It will facilitate comprehension.

Following the reviewer's comment, we described the legends for these figures.

Round 2

Reviewer 1 Report

The authors successfully addressed my comments, and therefore I fully recommend their revised manuscript for publication.